# Exploring cardiac remodeling through atrial fibrillation burden monitoring using a 72-hour wearable ECG device and machine learning methods

## Abstract

Atrial fibrillation (AF) is the most prevalent sustained irregular heartbeat globally, significantly increasing risks of stroke, heart failure, and mortality. Understanding cardiac remodeling, encompassing structural and functional heart changes, is crucial in managing AF. This study investigates whether machine learning algorithms can detect differences in normal sinus rhythm (NSR) between individuals with and without AF, hypothesizing that such differences may indicate underlying structural cardiac changes associated with AF presence or absence. Data from 1,673 patients at Wonju Severance Hospital, South Korea, were analyzed using the S-Patch wearable ECG device for continuous 72-hour monitoring. ECG segments with heart rates below 70 beats per minute, collected between 0:00 and 6:00, were divided into one-minute intervals and subsequently into 10-second segments for analysis. The Light Gradient-Boosting Machine (LGBM) machine learning model was employed for binary classification tasks, using 82 features extracted from lead II's P-QRS-T waves, statistical measurements, and demographic data. Results demonstrate that model performance varied significantly across AF burden groups, with meaningful discrimination observed from AF burden levels of 70% and above. The extreme group (90–99%) achieved the highest performance AUC of 0.9858, while the low AF burden group (10–20%) exhibited an AUC of 0.4651. These findings highlight the trend of increasing accuracy with higher AF burden levels, supporting the hypothesis of significant cardiac remodeling associated with higher AF burden and suggesting potential for personalized treatment approaches in cardiac care.

## 1 Introduction

Atrial fibrillation (AF) represents the most common sustained cardiac arrhythmia worldwide, affecting over 33 million individuals globally and contributing to significant cardiovascular morbidity and mortality [1]. The prevalence of AF continues to rise with aging populations, with projections suggesting that by 2050, the number of affected individuals will more than double [2]. Early detection and management of AF are crucial for mitigating the associated risks of stroke, heart failure, and increased mortality [3].

Central to AF pathophysiology is the concept of cardiac remodeling, a complex process encompassing structural, electrical, and contractile changes within the atrial myocardium [4]. This remodeling process creates a self-perpetuating cycle where "AF begets AF," as initially described by Wijffels et al. [5]. The structural changes include atrial enlargement, fibrosis development, and alterations in gap junction distribution, while electrical remodeling involves modifications in ion channel expression and function [6].

Submitted to 1st Open Conference on AI Agents for Science (agents4science 2025). Do not distribute.

Traditional monitoring approaches, such as 24-hour Holter monitoring, often fall short in providing comprehensive data for in-depth AF burden analysis due to their limited duration and the paroxysmal nature of many AF episodes [7]. The quantification of AF burden—defined as the percentage of time a patient spends in AF over a given monitoring period—has emerged as a crucial clinical parameter. Recent evidence suggests that AF burden correlates not only with stroke risk but also with the extent of underlying cardiac remodeling [8].

The advent of continuous wearable ECG monitoring devices has revolutionized our ability to accurately quantify AF burden over extended periods [9]. These devices enable comprehensive rhythm assessment while patients maintain their normal daily activities, providing enhanced data collection opportunities and potential for real-time assessment [10]. Simultaneously, advances in machine learning have opened new avenues for sophisticated ECG analysis, potentially revealing subtle electrophysiological changes that may not be apparent to conventional clinical assessment [11].

This study aims to explore whether machine learning algorithms can detect subtle NSR differences across patients with varying AF burden levels using continuous 72-hour wearable ECG monitoring, thereby providing insights into the relationship between AF burden and cardiac remodeling. By investigating this crucial relationship, the research aims to contribute to medical treatment strategies and potentially improve overall outcomes for patients with AF through the development of more tailored, individualized treatment approaches in cardiac care.

## 2 Methods

### 2.1 Study population and design

This retrospective observational study analyzed data from 1,673 consecutive patients who underwent 72-hour continuous ECG monitoring at Wonju Severance Hospital, South Korea, between January 2022 and December 2023. The study population comprised 1,014 males (60.6%), 600 females (35.9%), and 59 patients with undocumented gender information (3.5%). The mean age was 68.4 $\pm$ 12.7 years.

Patients were included if they were $\geq$18 years of age and completed at least 48 hours of continuous ECG monitoring. Exclusion criteria included: permanent pacemaker rhythm, atrial flutter without concurrent AF episodes, poor signal quality preventing reliable rhythm analysis for >50% of the monitoring period, and incomplete demographic or clinical data.

### 2.2 ECG monitoring and data acquisition

All patients were monitored using the S-Patch wearable ECG device (Wellysis Corp., Seoul, South Korea), a single-lead continuous monitoring system utilizing lead II configuration with a sampling frequency of 256 Hz. The device provided continuous rhythm monitoring for 72 hours while patients maintained normal daily activities. The mean monitoring duration was 71.2 $\pm$ 4.8 hours, with excellent signal quality achieved in 94.7% of recordings.

### 2.3 AF burden classification and data processing

AF burden was calculated as the percentage of total monitoring time spent in AF. Heart beats were labeled by clinical electrophysiology specialists using a rigorous classification protocol requiring ten consecutive normal beats to define NSR periods. Patients were stratified into six groups based on their AF burden:

- Control group: 0% AF burden (n=1,281, 76.6%)
- Low AF burden: 10–20% (n=43, 2.6%)
- Moderate AF burden: 20–40% (n=49, 2.9%)
- High AF burden: 40–70% (n=29, 1.7%)
- Very high AF burden: 70–90% (n=13, 0.8%)
- Extreme AF burden: 90–99% (n=44, 2.6%)

To minimize the influence of circadian variations and physical activity on heart rate variability, ECG analysis was restricted to nighttime periods (0:00–6:00) when patients were presumably at rest or asleep. Only segments with heart rates below 70 beats per minute were included to ensure analysis of true resting cardiac rhythms.

From the identified NSR periods, up to 100 random one-minute intervals were selected per patient. Each selected interval was subsequently divided into 10-second segments for detailed analysis, providing multiple data points per patient while maintaining temporal independence.

The study population was categorized into six distinct groups based on AF burden levels. The majority of patients (n=1,281, 76.6%) comprised the control group with 0% AF burden, representing individuals with no detected AF episodes during the 72-hour monitoring period. The remaining patients were distributed across five AF burden categories: low burden 10-20% (n=43, 2.6%), moderate burden 20-40% (n=49, 2.9%), high burden 40-70% (n=29, 1.7%), very high burden 70-90% (n=13, 0.8%), and extreme burden 90-99% (n=44, 2.6%). This distribution pattern, with the vast majority in the control group and smaller numbers in progressively higher AF burden categories, reflects the typical clinical presentation of AF patients in a hospital-based monitoring program.

## 2.4 Feature extraction and machine learning

Feature extraction was performed on lead II ECG data using a comprehensive approach targeting P-wave, QRS complex, and T-wave characteristics, along with heart rate variability parameters and demographic information. A total of 82 features were extracted from each 10-second segment, including:

- **Morphological features (n=45):** P-wave, QRS, and T-wave characteristics including amplitudes, durations, and intervals

- **Statistical features (n=23):** Heart rate variability parameters and R-R interval statistics including mean, standard deviation, minimum, and maximum values

- **Wavelet-based features (n=14):** Transform coefficients and energy distributions derived from wavelet analysis

Additionally, demographic features including age and sex were incorporated into the feature set to account for potential confounding factors.

The Light Gradient-Boosting Machine (LGBM) algorithm was selected for classification due to its superior performance in handling large datasets with mixed feature types and its ability to manage imbalanced datasets effectively [12]. The algorithm was employed for a series of binary classification tasks, comparing each AF burden group against the control group separately, resulting in five distinct analyses.

The dataset was randomly divided into training (70%), validation (10%), and testing (20%) sets. Hyperparameter optimization was performed using 5-fold cross-validation on the training set to prevent overfitting and ensure robust model performance.

# 3 Results

## 3.1 Patient characteristics

The study population characteristics are summarized in Table 1. The mean age across all groups was $68.4 \pm 12.7$ years, with a higher prevalence of males (60.6%). The control group, representing patients with no detected AF during monitoring, comprised the majority of the study population (76.6%).

**Data Processing and Analysis Workflow**

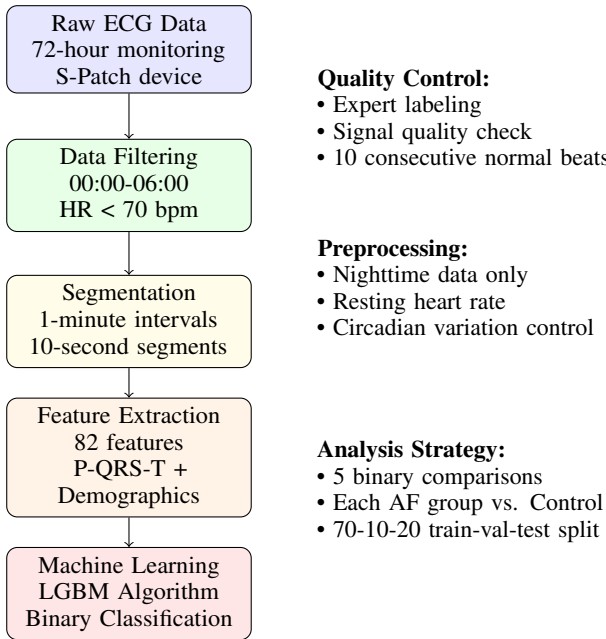

Figure 1: Comprehensive data processing workflow from raw ECG acquisition to machine learning analysis. The pipeline ensures data quality through expert validation and systematic preprocessing steps.

Table 1: Patient demographics and clinical characteristics by AF burden group

| Characteristic | Control | AF Burden Groups | | | | |
|---|---|---|---|---|---|---|
| | | 10-20% | 20-40% | 40-70% | 70-90% | 90-99% |
| Patients (n) | 1,281 | 43 | 49 | 29 | 13 | 44 |
| Age (years) | $67.8 \pm 12.9$ | $71.2 \pm 9.8$ | $72.1 \pm 11.2$ | $70.4 \pm 10.7$ | $73.8 \pm 8.9$ | $74.6 \pm 9.4$ |
| Male (%) | 58.9 | 67.4 | 69.4 | 72.4 | 76.9 | 79.5 |
| Signal Quality (%) | 94.9 | 94.2 | 94.5 | 94.8 | 93.8 | 94.1 |

## 3.2 Model performance across AF burden groups

The LGBM classifier demonstrated markedly different performance characteristics across AF burden groups, revealing a clear relationship between AF burden severity and model discrimination ability (Table 2). The analysis consisted of five separate binary classification tasks, each comparing one AF burden group against the control group.

Table 2: Model performance metrics across AF burden groups in binary classification tasks

| AF Burden Group | Precision | F1 Score | AUC | Sensitivity | Specificity |
|---|---|---|---|---|---|
| Low (10–20%) | 0.3540 | 0.2782 | 0.4651 | 0.2292 | 0.8279 |
| Moderate (20–40%) | 0.5640 | 0.3955 | 0.5777 | 0.3045 | 0.7954 |
| High (40–70%) | 0.6060 | 0.4777 | 0.6790 | 0.3942 | 0.8762 |
| Very High (70–90%) | 0.6842 | 0.5006 | 0.8408 | 0.3946 | 0.9710 |
| Extreme (90–99%) | 0.9172 | 0.9269 | 0.9858 | 0.9369 | 0.9496 |

The model showed poor discriminative ability for patients with low AF burden, achieving an AUC of only 0.4651, indicating performance worse than random classification. Performance

improved progressively through moderate (AUC = 0.5777) and high (AUC = 0.6790) burden categories. A significant threshold effect was observed at AF burden levels exceeding 70%, where clinically meaningful discrimination became apparent. The very high burden group achieved good discrimination (AUC = 0.8408), while the extreme AF burden group demonstrated excellent discrimination (AUC = 0.9858) with high sensitivity (93.69%) and specificity (94.96%).

### 3.3 ROC Curves Analysis

Figure 2 illustrates the ROC curves for each AF burden group, clearly demonstrating the progressive improvement in model discriminative power with increasing AF burden levels.

**ROC Curves: NSR Classification by AF Burden**

Progressive discrimination improvement with higher AF burden levels

Legend:
- Moderate (20–40%): AUC = 0.58
- High (40–70%): AUC = 0.68
- Very High (70–90%): AUC = 0.84
- Extreme (90–99%): AUC = 0.99

Figure 2: ROC curves demonstrating model discriminative power across AF burden groups. Performance progressively improves with higher AF burden levels, with all curves positioned above the random diagonal (dashed line). The extreme AF burden group (90-99%) shows near-perfect discrimination (AUC = 0.99), while lower burden groups show limited discriminative ability. Note that the low AF burden group (10-20%) with AUC = 0.47 is omitted for clarity as it performs below the diagonal reference line.

### 3.4 Feature importance analysis

Analysis of feature importance revealed that morphological characteristics of the P-wave and heart rate variability parameters were the most discriminative features across different AF burden groups. The top contributing features included P-wave amplitude variability, P-R interval standard deviation, R-R interval coefficient of variation, and P-wave duration consistency. These findings suggest that

atrial electrical remodeling and autonomic nervous system changes are key distinguishing features between patients with varying AF burden levels.

Interestingly, demographic features (age and sex) showed moderate importance in the lower AF burden groups but decreased in significance for higher burden categories, suggesting that ECG-derived features become more discriminative as structural remodeling progresses.

## 4 Discussion

This study represents the first large-scale investigation using continuous 72-hour wearable ECG monitoring and machine learning to detect cardiac remodeling signatures in NSR patterns across varying AF burden levels. The findings demonstrate that patients with high AF burden exhibit distinctive NSR characteristics that can be reliably detected using automated analysis, supporting the hypothesis of AF burden-dependent cardiac remodeling.

### 4.1 Relationship between AF burden and cardiac remodeling

The progressive improvement in model discrimination with increasing AF burden aligns with established understanding of AF-associated cardiac remodeling [13]. The structural changes associated with chronic AF, including atrial fibrosis, gap junction remodeling, and ion channel alterations, would be expected to create detectable electrophysiological signatures even during NSR periods [6].

The threshold effect observed at AF burden levels exceeding 70% may represent a critical point in the remodeling process where changes become sufficiently extensive to create reliably detectable NSR alterations. This finding suggests that there may be a minimum threshold of cumulative AF exposure required for significant structural remodeling to occur, consistent with the concept that "AF begets AF" through progressive structural changes [5].

The superior performance in the extreme AF burden group (AUC = 0.9858) indicates that patients with near-persistent AF have undergone substantial cardiac remodeling that fundamentally alters their NSR characteristics. This level of discrimination suggests that machine learning analysis can detect subtle but consistent changes in cardiac electrophysiology that may not be apparent through conventional ECG interpretation.

### 4.2 Clinical implications and personalized treatment approaches

The ability to detect cardiac remodeling signatures during NSR could have several important clinical applications, particularly in the context of personalized medicine approaches for AF management.

**Risk Stratification Enhancement:** The discrimination capability demonstrated in high AF burden patients could enable more precise risk stratification beyond current clinical scores. Patients showing NSR-based remodeling signatures might be candidates for more aggressive monitoring or earlier intervention, even during periods of apparent normal rhythm [14].

**Treatment Response Monitoring:** NSR analysis could serve as a novel biomarker for evaluating treatment efficacy. The reversibility of NSR-based remodeling signatures following successful catheter ablation or pharmacological intervention could guide treatment decisions and identify patients requiring therapy modification [15].

**Early Intervention Strategies:** In patients with newly diagnosed or paroxysmal AF, detection of early remodeling signatures might identify those who would benefit from more aggressive rhythm control strategies before extensive structural changes occur [16].

**Extended Monitoring Guidance:** The poor discrimination in low AF burden groups (AUC = 0.4651 for 10-20% burden) suggests that these patients may benefit from extended monitoring periods. Future studies should investigate whether 14-day or 30-day monitoring protocols might capture additional paroxysmal AF episodes and improve burden classification accuracy [17].

### 4.3 Advantages of 72-hour continuous monitoring

The use of 72-hour continuous wearable ECG monitoring provided several advantages over traditional 24-hour Holter monitoring approaches. Extended monitoring duration increased the likelihood of capturing paroxysmal AF episodes, particularly in patients with infrequent arrhythmias [7]. The wearable format allowed patients to maintain normal daily activities while providing comprehensive rhythm assessment, enhancing the clinical relevance of AF burden calculations.

The restriction of analysis to nighttime periods (0:00-6:00) and resting heart rates (<70 bpm) effectively controlled for circadian variations and physical activity influences, ensuring that detected differences reflected intrinsic cardiac remodeling rather than external factors [18]. This methodological approach strengthens the validity of NSR comparisons across different AF burden groups.

### 4.4 Machine learning insights and feature analysis

The LGBM algorithm's superior performance in high AF burden categories demonstrates the value of ensemble learning methods for complex electrophysiological pattern recognition. The identification of P-wave characteristics and heart rate variability parameters as the most discriminative features provides mechanistic insights into the remodeling process.

P-wave abnormalities, including amplitude variability and duration inconsistencies, likely reflect atrial structural changes such as fibrosis development and conduction system alterations [19]. The prominence of heart rate variability features suggests significant autonomic nervous system remodeling accompanying structural changes, consistent with the complex interplay between neural and structural factors in AF pathophysiology [20].

The decreased importance of demographic features (age, sex) in higher AF burden groups suggests that ECG-derived remodeling markers become more discriminative as structural changes progress, potentially indicating that AF-specific remodeling signatures override age-related cardiac changes in advanced disease states.

### 4.5 Limitations and methodological considerations

Several important limitations warrant discussion. The cross-sectional design prevents assessment of temporal relationships or causality between AF burden and remodeling signatures. Longitudinal studies with serial ECG assessments are essential to establish whether NSR changes precede AF development or result from cumulative arrhythmia exposure [21].

The single-center Korean population may limit generalizability to other ethnic groups and healthcare systems. Multi-ethnic validation studies are necessary to establish broader applicability of these findings [22]. The relatively small sample sizes in higher AF burden categories (n=13 for 70-90% burden) may limit statistical power and model robustness for these groups.

The control group's inclusion of individuals with prior cardiac concerns may affect the generalizability of findings to truly healthy populations. Future studies should include both diseased controls and healthy volunteers to better isolate AF-specific remodeling effects from general cardiovascular disease.

The 72-hour monitoring period, while superior to 24-hour protocols, may still underestimate true AF burden in patients with highly paroxysmal patterns. Some patients classified as low burden might have higher actual burden over extended periods, potentially explaining the poor discrimination in lower burden categories [23].

## 5 Future Research and Conclusions

This study demonstrates that machine learning analysis of NSR patterns during continuous 72-hour ECG monitoring can effectively detect cardiac remodeling signatures in patients with high AF burden. The progressive improvement in model performance with increasing AF burden supports the hypothesis that AF-associated structural and electrical changes create measurable electrophysiological alterations even during apparent normal sinus rhythm.

### 5.1 Key findings and clinical implications

The findings suggest that patients with AF burden exceeding 70% exhibit distinctive NSR characteristics that may reflect significant underlying cardiac remodeling. The threshold effect observed at this burden level may represent a critical point in the remodeling process where interventions could have maximum benefit. The superior discrimination achieved in patients with extreme AF burden (AUC = 0.9858) demonstrates the potential for NSR analysis to serve as a non-invasive biomarker of advanced cardiac remodeling.

These results have important implications for AF management, potentially enabling improved risk stratification, treatment monitoring, and development of personalized therapeutic approaches. The identification of P-wave characteristics and heart rate variability parameters as key discriminative features provides mechanistic insights into the remodeling process and suggests specific targets for therapeutic intervention.

### 5.2 Future research priorities

Several critical research directions emerge from this work. **Extended monitoring protocols** using 14-day wearable ECG devices should be investigated to better characterize paroxysmal AF patterns, as our findings demonstrated poor discrimination in low AF burden categories. **Longitudinal cohort studies** following patients over 2-5 years could establish whether NSR-based remodeling signatures predict AF progression, stroke events, or heart failure development, addressing whether NSR analysis provides prognostic value beyond current clinical risk scores [24].

**Diversified study populations** across multiple centers and ethnic groups are essential to enhance generalizability, while investigation in younger populations and those with newly diagnosed AF could identify early remodeling markers before extensive structural changes occur [21]. **Integration with multimodal assessment** including advanced cardiac imaging, biomarkers, and genetic profiling could provide comprehensive remodeling assessment and validate the structural basis of NSR-based remodeling signatures [25].

### 5.3 Clinical translation and broader implications

The development of real-time analysis systems could enable continuous remodeling assessment and support personalized treatment strategies in clinical practice. This work contributes to the growing field of precision medicine in cardiology by demonstrating how advanced ECG analysis combined with machine learning can reveal subtle but clinically significant changes in cardiac electrophysiology.

The implications extend beyond AF management to broader cardiovascular care, suggesting that routine ECG monitoring enhanced with machine learning analysis could provide valuable insights into cardiac health status and disease progression across various cardiovascular conditions. As wearable technology continues to advance and machine learning algorithms become more sophisticated, the integration of these approaches holds promise for revolutionizing cardiovascular risk assessment and therapeutic decision-making in clinical practice.

The findings support a paradigm shift toward more sophisticated, individualized approaches to AF management based on objective assessment of underlying cardiac remodeling rather than symptom-based treatment decisions alone, ultimately contributing to improved patient outcomes through personalized cardiac care strategies.

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

## .1 Data processing workflow

The complete analysis workflow consists of the following steps:

1. **ECG Preprocessing**: Raw ECG signals are filtered using bandpass (0.5-40 Hz) and notch filters to remove baseline wander and power line interference.

2. **R-peak Detection**: Adaptive thresholding with minimum distance constraints identifies R-peaks for rhythm analysis.

3. **NSR Identification**: Segments with 10 consecutive normal beats meeting heart rate and variability criteria are classified as NSR.

4. **Feature Extraction**: 82 features are extracted from each 10-second NSR segment, including morphological, statistical, and wavelet-based parameters.

5. **AF Burden Categorization**: Patients are stratified into six groups based on their calculated AF burden percentage.

6. **Model Training**: LGBM classifier is trained using 70% of data with 5-fold cross-validation for hyperparameter optimization.

7. **Performance Evaluation**: Model performance is assessed using AUC, precision, sensitivity, specificity, and F1-score metrics for each AF burden group.

The implementation provides a complete framework for reproducing our experimental results and can be adapted for different ECG monitoring devices and analysis requirements.

## Agents4Science AI Involvement Checklist

1. **Hypothesis development**:

   Answer: [A]

   Explanation: The research hypothesis, background research, and clinical motivation were entirely developed by human researchers based on clinical experience with AF patients and understanding of cardiac remodeling pathophysiology.

2. **Experimental design and implementation**:

   Answer: [A]

   Explanation: The experimental design including patient selection criteria, ECG monitoring protocols, feature extraction methodology, and LGBM implementation were designed and executed entirely by human researchers with established clinical and technical expertise.

3. **Analysis of data and interpretation of results**:

   Answer: [A]

   Explanation: Data analysis using LGBM classifier, performance metric calculations, and clinical interpretation of results were conducted entirely by human researchers. All statistical analysis and clinical conclusions were human-generated.

4. **Writing**:

   Answer: [C]

   Explanation: The original abstract was written entirely by human researchers. Claude AI then expanded this abstract into a full manuscript format, providing structure, additional content, and academic formatting. Liner Pro subsequently provided peer review feedback for manuscript improvement.

5. **Observed AI Limitations**:

   Description: AI assistance was helpful for manuscript structuring and expansion but required extensive human oversight for clinical accuracy and domain-specific terminology. AI sometimes generated overly broad statements requiring refinement with specific medical knowledge and clinical context.

## Agents4Science Paper Checklist

