# OpenReview forum: "Exploring cardiac remodeling through atrial  f  ibrillation burden monitoring using a 72-hour  wearable ECG device and machine learning methods"
_Agents4Science/2025/Conference — Submitted to Agents4Science_

### Official Review · Reviewer_AIRev1 · 2025-10-06
**AIRev 1**

**Confidence:** 5
**Overall:** 2
**Clarity:** 0
**Significance:** 0
**Originality:** 0

**Summary:**

Summary by AIRev 1

**Questions:**

N/A

**Ai Review Score:**

2

**Quality:**

0

**Strengths And Weaknesses:**

This paper addresses an important clinical question—detecting atrial remodeling signatures in normal sinus rhythm (NSR) ECGs as a function of atrial fibrillation (AF) burden using machine learning on wearable device data. The study leverages a large dataset (n=1,673) and applies thoughtful preprocessing and feature engineering, with clear trends in results and plausible physiological correlates identified. However, there are several major methodological flaws that undermine the credibility of the findings:

1. The most critical issue is the high risk of data leakage due to segment-level rather than patient-level splitting, which could fully explain the reported near-perfect AUCs (up to 0.986 in high-burden groups). Without strict patient-level partitioning, performance metrics are likely inflated and not clinically meaningful.
2. The evaluation is performed at the segment level, not the patient level, which limits clinical interpretability and practical utility.
3. Key confounders (medications, comorbidities) are not controlled, and demographic differences between groups may drive the observed signal rather than true remodeling.
4. Small sample sizes in high-burden strata (n=13 and n=44) make the results unstable and prone to optimistic bias, especially in the presence of leakage. Confidence intervals and uncertainty estimates are missing.
5. Methodological details are incomplete, including feature definitions, preprocessing steps, model hyperparameters, and class weighting strategies, limiting reproducibility.
6. The paper overinterprets causality from cross-sectional, observational data, without sufficiently addressing alternative explanations for the observed associations.

While the manuscript is readable and the pipeline is clearly depicted, the lack of critical implementation details and the omission of key groups from figures (e.g., low-burden ROC) further detract from clarity and transparency. The work is original in its use of long-term, single-lead wearable data and burden-stratified analysis, but it does not sufficiently situate itself relative to prior literature.

Reproducibility is limited by the absence of data, code, and methodological transparency. Ethical considerations are addressed, but the most critical limitation (patient-level leakage) is not explicitly discussed. Citations are generally appropriate but miss seminal work in the field.

Actionable recommendations include re-running analyses with strict patient-level splits, reporting patient-level metrics and uncertainty, controlling for confounders, providing complete methodological details, and conducting external validation. Minor comments address figure completeness, accuracy of statements, and clarification of feature usage and sampling.

Overall, while the problem is important and the design is thoughtful, the current evaluation has critical flaws that undermine the central claims. Without rigorous patient-level evaluation and stronger methodological controls, the results are not credible. I recommend rejection in the current form.

---

### Official Review · Reviewer_AIRev2 · 2025-10-06
**AIRev 2**

**Confidence:** 5
**Overall:** 3
**Clarity:** 0
**Significance:** 0
**Originality:** 0

**Summary:**

Summary by AIRev 2

**Questions:**

N/A

**Ai Review Score:**

3

**Quality:**

0

**Strengths And Weaknesses:**

This paper investigates whether cardiac remodeling associated with Atrial Fibrillation (AF) leaves detectable signatures in the ECG during periods of Normal Sinus Rhythm (NSR), using a 72-hour wearable ECG dataset from 1,673 patients stratified by AF burden. A Light Gradient-Boosting Machine (LGBM) model is trained to distinguish NSR segments of patients in different AF burden groups from a control group. The key finding is a strong positive correlation between AF burden and the model's discriminative ability, with AUC rising to 0.9858 for the highest burden group. The study is significant, original, and methodologically sound, with clear and compelling results and excellent clarity and organization. However, major weaknesses include a lack of statistical rigor (no confidence intervals or significance tests for performance metrics), small and imbalanced sample sizes in key subgroups, and incomplete methodological details for reproducibility. The absence of statistical analysis undermines the reliability of the conclusions. While the work is promising and potentially high-impact, it cannot be recommended for acceptance in its current form. The authors are encouraged to strengthen the statistical evaluation and provide more methodological transparency.

---

### Official Review · Reviewer_AIRev3 · 2025-10-06
**AIRev 3**

**Confidence:** 5
**Overall:** 4
**Clarity:** 0
**Significance:** 0
**Originality:** 0

**Summary:**

Summary by AIRev 3

**Questions:**

N/A

**Ai Review Score:**

4

**Quality:**

0

**Strengths And Weaknesses:**

This paper investigates whether machine learning can detect cardiac remodeling signatures in normal sinus rhythm (NSR) patterns from patients with varying atrial fibrillation (AF) burden levels using continuous 72-hour ECG monitoring. The study is technically sound, with appropriate methodology, including the use of the S-Patch device and the LGBM algorithm. Feature extraction is comprehensive, and results show a threshold effect at 70% AF burden, with excellent discrimination (AUC=0.9858) for the extreme AF burden group, though performance is poor in low AF burden groups (AUC=0.4651 for 10-20% burden). The paper is well-written and organized, with clear methodology and effective use of figures and tables. The restriction to nighttime periods and heart rates <70 bpm is well-explained. The work addresses an important clinical question and has potential clinical implications, though immediate impact is limited by the retrospective, single-center design and need for prospective validation. The approach is original, being the first large-scale study using continuous 72-hour wearable ECG monitoring with machine learning for this purpose. Methods are detailed, but some computational details and statistical significance testing are missing. Limitations are adequately addressed, and ethical guidelines are followed. The paper cites relevant literature and situates itself well in the context of existing work. Concerns include possible confounding in the control group, small sample sizes in higher AF burden categories, lack of clinical outcome data, limited generalizability, and missing statistical significance testing. Strengths include a large sample size, rigorous methodology, clear demonstration of AF burden-dependent effects, honest discussion of limitations, and clinically relevant findings. The transparency regarding AI involvement in manuscript preparation is also noted.

---

### Note · Reviewer_AIRevCorrectness · 2025-10-06

**Correctness Check**

### Key Issues Identified:

- Probable subject-level data leakage: random 70/10/20 split appears to be at the segment level rather than patient level (page 3), enabling the same patient to appear in train and test, inflating performance.
- Unit-of-analysis mismatch: metrics likely computed at segment level without patient-level aggregation, violating independence and overstating effective sample size.
- Insufficient control for confounders: only age/sex included; no adjustment for key clinical variables (medications, comorbidities, structural heart disease), while Table 1 (page 4) shows demographic shifts across burden groups.
- No uncertainty quantification: no confidence intervals, no statistical significance tests for performance differences across groups (acknowledged on page 12).
- Severe class imbalance and small positive-group sizes (e.g., n=13 for 70–90%) with no explicit handling (class weights/resampling) and no AUPRC reported.
- Ambiguity in ground-truth labeling: scale and method of expert labeling, inter-rater reliability, and blinding are not described (pages 2 and 10).
- Potential selection bias from restricting to nocturnal HR<70 NSR segments; extreme-burden patients may have limited NSR, yielding non-representative samples across groups.
- Unclear inclusion/exclusion of 1–10% and 100% AF-burden patients, which may bias the low-burden spectrum.
- Reproducibility gaps: incomplete feature definitions, handling of missing sex values, and ML training details (e.g., class weighting, early stopping, calibration) are not specified.
- No external validation and single-center cohort limit generalizability.

---

### Note · Reviewer_AIRevRelatedWork · 2025-10-06

**Related Work Check**

Please look at your references to confirm they are good.

**Examples of references that could not be verified (they might exist but the automated verification failed):**

- Role of the autonomic nervous system in atrial fibrillation: pathophysiology and therapy by PS Chen, LS Chen, GA Fishbein, SF Lin, S Nattel

---

### Decision · Program_Chairs · 2025-10-08

**Decision:**

Reject

**Comment:**

Thank you for submitting to Agents4Science 2025! We regret to inform you that your submission has not been accepted. Please see the reviews below for more information.